# Search for Indexes to Evaluate Trends in Antibiotic Use in the Sub-Prefectural Regions Using the National Database of Health Insurance Claims and Specific Health Checkups of Japan

**DOI:** 10.3390/antibiotics11060763

**Published:** 2022-06-02

**Authors:** Kanako Mizuno, Ryo Inose, Yuna Matsui, Mai Takata, Daisuke Yamasaki, Yoshiki Kusama, Ryuji Koizumi, Masahiro Ishikane, Masaki Tanabe, Hiroki Ohge, Norio Ohmagari, Yuichi Muraki

**Affiliations:** 1Department of Clinical Pharmacoepidemiology, Kyoto Pharmaceutical University, Kyoto 607-8414, Japan; ky17326@ms.kyoto-phu.ac.jp (K.M.); inose2019@mb.kyoto-phu.ac.jp (R.I.); ky17306@ms.kyoto-phu.ac.jp (Y.M.); ky17187@ms.kyoto-phu.ac.jp (M.T.); 2Department of Infection Control and Prevention, Mie University Hospital, Tsu 514-8507, Japan; yamadai@med.mie-u.ac.jp (D.Y.); m-tanabe@clin.medic.mie-u.ac.jp (M.T.); 3Division of General Pediatrics, Department of Pediatrics, Hyogo Prefectural Amagasaki General Medical Center, Amagasakic 660-8550, Japan; stone.bagle@gmail.com; 4AMR Clinical Reference Center, Disease Control and Prevention Center, National Center for Global Health and Medicine, Tokyo 162-8655, Japan; rykoizumi@hosp.ncgm.go.jp (R.K.); mishikane@hosp.ncgm.go.jp (M.I.); nohmagari@hosp.ncgm.go.jp (N.O.); 5Department of Infectious Diseases, Hiroshima University Hospital, Hiroshima 734-8551, Japan; ohge@hiroshima-u.ac.jp

**Keywords:** National Database of Health Insurance Claims and Specific Health Checkups, antimicrobial use, DDDs/1000 inhabitants per day, DOTs/1000 inhabitants per day, Patients/1000 inhabitants/day, secondary medical area

## Abstract

The evaluation indexes of antimicrobial use (AMU) in sub-prefectural regions have not been established because these regional units are susceptible to the effects of population inflows and outflows. We defined the difference in AMU calculated each year as a new evaluation index and compared the AMU of secondary medical areas with those already reported for Japan and each prefecture. Patients/1000 inhabitants/day (PID) for oral antibiotics in 2013 and 2016 were calculated using the National Database of Health Insurance Claims and Specific Health Checkups. ΔPID was defined as the difference between the PIDs in 2013 and 2016. Differences in AMUs for Japan and prefectures that have already been published were also calculated, and the concordance rate with ΔPID in each secondary medical area was evaluated. Antibiotics and age groups with less than 50% concordance between secondary medical area and previously reported AMU changes were observed. This revealed that even at the secondary medical area level, which is more detailed than the prefectural level, the AMU changes were not consistent. Therefore, in order to appropriately promote measures against antimicrobial resistance, we suggest the necessity of not only surveying AMU at the national or prefectural levels but also examining sub-prefectural trends in AMU.

## 1. Introduction

The emergence of drug-resistant bacteria due to inappropriate use of antibiotics is a worldwide concern [1,2]. Drug-resistant bacteria affect the treatment and prevention of infectious diseases and are a threat to human life. It is reported that if no action is taken, antimicrobial resistance (AMR) could kill 10 million people a year by 2050 and cause significant economic losses [2]. To resolve issues related to AMR, the World Health Organization (WHO) adopted a Global Action Plan on AMR in May 2015, requesting all member states to develop a national action plan on AMR within two years [3].

Japan developed a National Action Plan on AMR in 2016, consisting of six strategies [4]. One of the strategies is the establishment of an antimicrobial use (AMU) surveillance system. AMU is known to be closely associated with the emergence of drug-resistant bacteria [4,5]. Therefore, it is important to provide clear feedback on AMU trends with the implementation of AMR measures.

In Japan, nationwide AMUs, and those by prefecture have been clarified using sales data or the National Database of Health Insurance Claims and Specific Health Checkups (NDB) [6,7,8,9]. The NDB is constructed with electronic claims data based on actual prescriptions and does not include dead stock or disposed of items within medical institutions. Therefore, the NDB is considered to be a database that more accurately reflects drug use than sales volume information [10]. To date, AMUs based on the usage in Japan and in each prefecture based on the NDB from 2013 to 2019 have been reported, revealing that AMUs in Japan increased until 2016 and then decreased [6]. On the other hand, some prefectures have reported that their trends in AMU differ from the trend in Japan as a whole [7]. Thus, it is necessary to evaluate AMU in each region and take countermeasures.

In Japan, there is a regional unit called a “medical area”, which was established prefecture-wise, in accordance with the Medical Care Act [11]. These medical areas are divided into three geographical levels: primary, secondary, and tertiary medical regions. Primary medical areas consist of municipalities as units, capable of providing basic primary outpatient medical care. Secondary medical areas consist of several primary medical areas designed to provide medical care related to general hospitalization, including emergency care. Tertiary medical areas consist mainly of prefectures that are capable of providing advanced medical care requiring specialized skills. The most frequently used basic unit for healthcare planning in Japan is the secondary medical area. In addition, although a secondary medical area is an important unit for establishing medical care in a region, such as a medical institution cooperation, the trend of AMUs in secondary medical areas has not been clarified.

The WHO and the Centers for Disease Control and Prevention recommend using the dose and administration period to evaluate AMU [12]. DDDs/1000 inhabitants per day (DID) based on the dose and DOTs/1000 inhabitants per day (DOTID) based on the administration period are commonly used. However, the dose and the administration period depend on the patient’s background and type of infection. Therefore, we reported that Patients/1000 inhabitants/day (PID), based on the number of patients who were administered with antibiotics, is a more useful index for evaluating AMU in each region than DID or DOTID [13].

The population used to correct AMU is an issue when calculating AMU using the NDB. There are two types of the population used for correction: the population based on the residential area (“nighttime population”) and the population based on one’s workplace or activity (“daytime population”) [14]. Previously, we have shown that the DID corrected for the nighttime population is greater than the DID corrected for the daytime population in urban areas [15]. A factor responsible for this during the daytime is that people gather in urban areas to commute to work or school. Therefore, detailed regional units such as secondary medical areas are susceptible to the effects of population inflows and outflows. This makes it difficult to evaluate AMUs simply by calculating them using conventional indicators.

This study aimed to explore evaluation indexes for AMR measures in regions more detailed than prefectures. Therefore, we defined the difference in AMU calculated in each year as a new evaluation index of AMU and compared this index of a secondary medical area with those already reported for Japan and each prefecture.

## 2. Methods and Materials

### 2.1. Study Design

The survey period was set to 2013, which is the reference year for the National Action Plan on AMR, and 2016, which is the year the National Action Plan on AMR was implemented. The study areas were the six secondary medical areas in the Kyoto Prefecture (Tango, Chutan, Nantan, Kyoto-Otokuni, Yamashiro-kita, and Yamashiro-minami). Table 1 shows the population and number of hospitals and clinics in each secondary medical area. In Japan, a city is defined as a designated city if it has a population of 500,000 or more and is designated by cabinet order [16]. Therefore, in this study, each secondary medical area was classified according to whether it had a population of more or less than 500,000.

The antibiotics surveyed were oral third-generation cephalosporins (J01DD), oral quinolones (J01MA, J01MB), oral macrolides (J01FA), and all oral antibiotics based on the fourth level of the Anatomical Therapeutic Chemical (ATC) classification system as defined by WHO [17]. The drugs included in the oral antibiotics were combinations for eradication of Helicobacter pylori (A02BD), other intestinal anti-infectives (A07AA), antiprotozoals (P01AB), and systemic antibiotics (J01) based on the second level of the ATC classification system. Antiviral and antifungal agents were not included in this study. The target age groups were children (0–14 years), working-age individuals (15–64 years), the elderly (65 years and older), and all ages.

**Table 1 antibiotics-11-00763-t001:** Characteristics in each secondary medical area in the Kyoto prefecture.

Classification *^1^	Secondary Medical Area	Populations *^2^	Number of Hospitals *^3^	Number of Clinics *^3^
Big city	Kyoto-Otokuni	1,555,461	106	1717
Small city-1	Yamashiro-kita	433,858	23	303
Small city-2	Chutan	190,822	17	162
Small city-3	Nantan	132,537	10	101
Small city-4	Yamashiro-minami	123,789	3	92
Small city-5	Tango	94,142	6	76

*^1^: A population of 500,000 or more was defined as a “Big city”, while a population of less than 500,000 was defined as a “Small city”; *^2^: Data from 2021 was used for the population [18]; *^3^: The number of healthcare facilities used data from 2019 [18].

### 2.2. Data Source

In this study, data were obtained from the NDB. The NDB has anonymized health insurance claims data and data on specific health checkups, along with specific health guidance. The NDB includes electronic health insurance claims data for medical, diagnosis procedure combination, dental care, and dispensed drugs; it does not include paper claims data. The use of NDB in this study was approved by the Ministry of Health, Labour and Welfare.

Population by secondary medical area was obtained from the portal site of official statistics of Japan (e-stat) [18], based on the basic resident register population, by municipalities by age group. These populations were each aggregated by secondary medical area and defined as the nighttime population. AMUs for Japan and prefectures were compiled from the 2013 and 2016 NDBs published by the AMR Clinical Reference Center (AMRCRC) [19]. This value was compared to the AMU of each secondary medical area.

### 2.3. Calculation of AMU Based on NDB and Evaluation for Trends of AMU

AMU was calculated as PID using the formula shown below (1). In this study, PID was calculated by dividing the number of patients administrated into inpatients (medical inpatient receipts) and outpatients (medical outpatient receipts and dispensed receipts) [13]. The number of patients administered with antibiotics was obtained from the NDB. If this number was less than 10, it was treated as 0. 

To evaluate trends of AMU from 2013 to 2016, PIDs for each target oral antibiotic in each age group were categorized by secondary medical area and the categories of inpatient and outpatient. ΔPID was defined as the difference between the PIDs in 2013 and 2016 (2). The amplitude of ΔPID in the categories of inpatient and outpatient was calculated by determining their respective differences between the maximum and minimum values. Aggregate values published by the AMRCRC were also determined using the same method of differences, with a value of less than 0 defined as a “decrease” and all other values as an “increase” (Table 2). The concordance rate was calculated between the ΔPID of each secondary medical area and the value in Table 2.
PID (Patients/1000 inhabitants/day) = Number of patients administered with antibiotics/(nighttime population/1000 inhabitants)/365 (days)(1)
ΔPID = (PID in 2016) − (PID in 2013)(2)

### 2.4. Statistical Analysis and Ethical Considerations

Bee-swarm plots of ΔPID in the inpatient and outpatient categories were prepared using JMP^®^ Pro 16 (SAS Institute Inc., Cary, NC, USA). The same method was also used for sensitivity analysis in secondary medical areas in other prefectures (Hiroshima and Mie prefectures) (Appendix A). The study was approved by the Ethics Committee of Kyoto Pharmaceutical University (approval number: 20–25).

## 3. Result

### 3.1. Trends of AMUs in Secondary Medical Areas in the Kyoto Prefecture from 2013 to 2016

Trends of AMUs in the Kyoto secondary medical area from 2013 to 2016 are shown in Table 3. Regardless of the type of antibiotic and age groups, the increase and decrease in AMU differed by region. In addition, the amplitude of ΔPID in outpatients was 9.91 times greater than that in inpatients. The distribution of ΔPID for inpatients and outpatients in the Kyoto prefecture is shown in Figure 1. The ΔPID of inpatients was concentrated around 0, while the ΔPID of outpatients was widely dispersed.

As a sensitivity analysis, the same study was conducted in secondary medical areas in the Mie and the Hiroshima prefectures. Similarly, in these two prefectures, the increase and decrease in AMU differed by region regardless of the type of antibiotics and age groups. The amplitudes of ΔPID for outpatients in the Mie and the Hiroshima prefectures were 17.62 and 38.84 times those for inpatients, respectively (Appendix A). As in Kyoto, the ΔPID for inpatients was concentrated around 0, while the ΔPID for outpatients was widely dispersed (Appendix A).

### 3.2. Relationship between Published Values in Japan or the Kyoto Prefecture and AMU Trends in Each Secondary Medical Area in the Kyoto Prefecture

Table 4 shows the concordance rate between the published values of Japan/the Kyoto prefecture and the trend of AMU in each secondary medical area in the Kyoto prefecture. For each antibiotic group and each age group, none of the trends in AMU in the secondary medical area in the Kyoto prefecture matched those previously reported. In addition, several antibiotic and age groups with less than 50% concordance rate of change between secondary medical areas and Japanese AMUs were observed. The concordance rates in the Hiroshima and the Mie prefectures also differed as in the Kyoto prefecture (Appendix A).

## 4. Discussion

This study revealed that trends of oral antibiotic use in secondary medical areas from 2013 to 2016 differed from Japan and the prefectural trends. This suggests the need to identify AMUs at a sub-prefectural level in addition to these primary trends to properly promote AMR countermeasures. It was also inferred that it may be useful to use differential rather than simple values to evaluate AMUs in the sub-prefectural areas.

The amplitude of ΔPID for outpatients in the Kyoto prefecture was higher than that for inpatients and was similar in other prefectures (Figure 1, Appendix A). Therefore, it is conceivable that there was little change in the number of patients using oral antibiotics in inpatients more than in outpatients, regardless of age or antibiotic. This might be because injectable antibiotics are primarily used in inpatients [10]. In recent years, various interventions have been implemented for the use of oral antibiotics in hospitalized patients [20,21,22]. However, no such reports were observed before 2016, when the National Action Plan on AMR was developed, suggesting that the number of patients using oral antibiotics in hospitalized patients scarcely changed. On the other hand, because the ΔPID in outpatients had been dispersed, it was found that outpatients had variability in oral antibiotic use. Therefore, it was considered necessary to take appropriate AMR measures separately for outpatients and inpatients.

In the outpatient group, an increasing trend was observed for targeted drugs in the National Action Plan on AMR in children (Table 3, Appendix A). In Japan, medical reimbursement was introduced in 2018 to allow reimbursement for not prescribing antibiotics for children under 3 years of age with upper respiratory tract infections [23]. The background for the development of this medical reimbursement may have been influenced by the increase in AMU in children identified in this study. On the other hand, a decrease in the number of drugs targeted by the National Action Plan on AMR in children was observed in some regions (Table 3, Appendix A). Since the types of antibiotics frequently used were reported to vary by hospital, clinic, and clinical department [24], it was inferred that sub-prefectural units would be influenced by the hospitals, clinics, and departments included in the scope of coverage.

Previous reports indicated that AMU variation differs between Japan and prefectures [7]. As one of these factors, the inflow and outflow of the prefectural population may have affected the AMU [15]. This study revealed that the variation in AMU is not consistent even at the regional level, such as secondary medical areas, which are more detailed than prefectures (Table 4, Appendix A). The National Action Plan on AMR in 2016 has an outcome goal of reducing the number of AMUs nationwide by 50% [4]. Other countries have also set similar numerical targets [25]. However, it was inferred that even if the outcome goals are based on national AMU values, they may differ from the actual situation in the sub-prefectural regions. Therefore, it was suggested that in order to appropriately promote AMR measures, it is necessary to not only survey AMUs at the national or prefectural levels, but also to clarify the trends of AMUs at the sub-prefectural levels.

A commonly used index of AMU including PID is calculated by dividing the number of patients administered with antibiotics, the dose, and the administration period, by the total population of that region. This study was conducted in a secondary medical area, an area that is affected by the inflow and outflow of the population between regions [15]. Therefore, the difference in AMU values from 2013 to 2016 was used as an evaluation indicator. Evaluation using differences in AMU during the study period is a useful method because it can be applied in the evaluation index of AMU other than PID, and can reflect the antibiotic selection pressure after eliminating the effects of population inflows and outflows. In addition, this method makes it possible to evaluate whether the reduction in AMU has reached the goals of the guidelines in each country.

Differences in AMUs at the regional level have been reported in other countries, such as the United States and Germany [26,27,28,29]. In both cases, regional units are defined according to the situation in each country so that the effectiveness of AMR measures can be more effectively evaluated. This study revealed for the first time, the trend of AMUs in secondary medical areas, the basic unit of medical planning in Japan, which differs from the trend in Japan and the prefecture. To make comparisons between countries, it is necessary to understand trends in AMUs at the national levels, but there is also a need to understand trends in AMUs at sub-prefectural levels, such as secondary medical areas, to evaluate the effectiveness of AMR measures.

This study has several limitations. First, because of the use of the NDB, services not included in the data, such as public funding for specific pediatric chronic diseases and public assistance, cannot be considered. However, the NDB used in this study covers more than 97% of claims in Japan and is considered to reflect the actual situation in Japan [30]. Second, we were unable to assess the association of confounders with variation in AMU due to the unavailability of published information related to secondary medical areas in the survey years. Therefore, it is necessary to develop an environment in which a variety of healthcare-related information can be aggregated over time and made available for secondary use in sub-prefectural regions. However, even with these limitations, the ΔPID identified in this study is a useful tool for evaluating AMU variation in the sub-prefectural areas and will contribute to the promotion of AMR measures worldwide.

## 5. Conclusions

This study revealed that trends of oral antibiotic use from 2013 to 2016 in secondary medical areas differed from the Japan and prefectural trends. The method used in this study can be applied not only to Japan but also to other regions of the world and is expected to be a useful source of information in promoting AMR countermeasures. Future studies should continue to evaluate trends of antibiotic use in sub-prefectural regions and their relationship to confounding factors such as population, number of medical facilities, and number of health care workers.

## Figures and Tables

**Figure 1 antibiotics-11-00763-f001:**
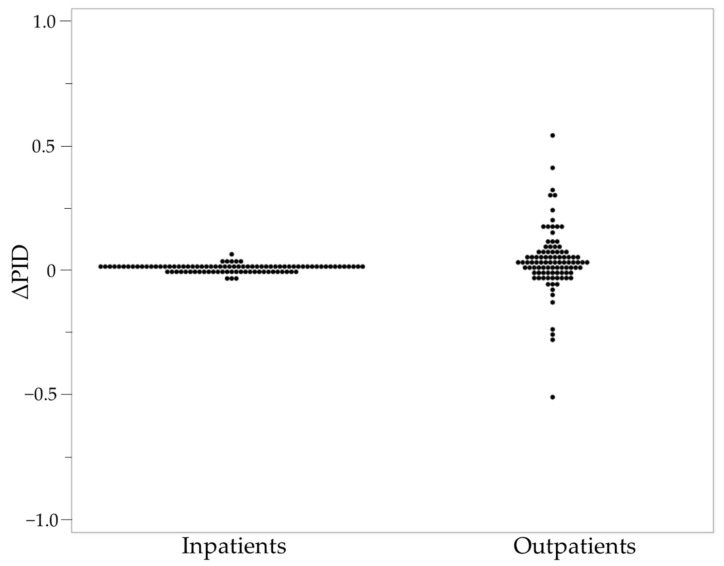
Bee-swarm plots of the ΔPID for inpatients and outpatients in the Kyoto prefecture. The left axis shows the ΔPID of each secondary medical area. The lower axis shows the number of inpatients and outpatients. PID: Patients/1000 inhabitants/day.

**Table 2 antibiotics-11-00763-t002:** Changes of previously reported AMU [19].

	Japan				Kyoto			
	<15 Years	15–64 Years	>64 Years	All Ages	<15 Years	15–64 Years	>64 Years	All Ages
**Third-generation cephalosporins**	↓	↑	↑	↑	↓	↑	↑	↑
**Quinolones**	↑	↑	↓	↑	↑	↑	↓	↑
**Macrolides**	↑	↑	↓	↑	↑	↑	↑	↑
**Total**	↑	↑	↑	↑	↑	↑	↑	↑

The increase is indicated by “↑” and the decrease by “↓”. The gray areas show the antibiotics and age groups with decreased AMU. AMU: Antimicrobial use.

**Table 3 antibiotics-11-00763-t003:** Changes in oral antibiotic use stratified by age group and secondary medical area in the Kyoto prefecture from 2013 to 2016.

	Inpatients				Outpatients			
Third−Generation Cephalosporins	<15 Years	15−64 Years	>64 Years	All Ages	<15 Years	15−64 Years	>64 Years	All Ages
Big city	−0.0066	−0.0074	−0.025	−0.013	−0.017	0.018	0.074	0.000029
Small city−1	0.0034	0.0030	0.017	0.0059	0.020	0.023	0.054	0.0083
Small city−2	−0.0046	−0.00058	0.013	−0.00099	−0.28	−0.044	0.027	−0.086
Small city−3	−0.0089	−0.0029	−0.034	−0.013	−0.25	0.024	0.11	−0.018
Small city−4	−0.00074	−0.0024	0.00074	−0.0022	−0.10	−0.0053	0.061	−0.029
Small city−5	−0.0047	−0.0088	−0.0084	−0.011	−0.51	−0.065	−0.025	−0.13
**Quinolones**								
Big city	0.00062	−0.0019	−0.0013	−0.0028	0.089	0.041	0.065	0.035
Small city−1	0.0021	0.00090	0.0040	0.0017	0.039	0.012	0.053	0.016
Small city−2	−0.0023	−0.0012	0.0045	−0.0017	0.0094	0.0021	0.039	−0.0037
Small city−3	0.00	−0.00081	−0.0034	−0.0026	0.027	0.026	0.054	0.021
Small city−4	0.0037	−0.00043	−0.022	−0.0045	0.018	−0.0091	0.0012	−0.016
Small city−5	−0.0028	−0.00051	−0.0078	−0.0047	0.00012	−0.053	−0.025	−0.054
**Macrolides**								
Big city	−0.0019	−0.00044	−0.0019	−0.0015	0.11	0.082	0.097	0.067
Small city−1	0.0025	0.0024	0.0066	0.0033	0.18	0.054	0.062	0.059
Small city−2	0.0058	0.00077	0.0066	0.0020	0.021	0.052	0.11	0.036
Small city−3	0.0053	0.00074	0.0020	0.0012	0.091	−0.0040	0.044	0.0016
Small city−4	0.0086	0.00029	−0.017	−0.0022	0.17	0.027	−0.015	0.024
Small city−5	−0.0030	−0.0014	0.0029	−0.0012	−0.26	0.025	0.062	−0.023
**Total**								
Big city	−0.0041	−0.0077	−0.0028	−0.012	0.41	0.18	0.32	0.16
Small city−1	0.019	0.013	0.063	0.025	0.54	0.15	0.24	0.17
Small city−2	0.013	−0.00040	0.039	0.0030	−0.031	0.047	0.30	0.013
Small city−3	0.0088	0.0036	−0.0087	−0.0030	−0.034	0.072	0.30	0.048
Small city−4	0.022	−0.0019	−0.043	−0.0081	0.042	0.032	0.037	−0.028
Small city−5	0.018	−0.0061	0.025	−0.0034	−0.037	0.0061	0.20	−0.037

The values show the ΔPID, which is the difference between the PIDs in 2013 and 2016. The gray areas show the regions and age groups with decreased AMU. A population of 500,000 or more was defined as a “Big city”, while a population of less than 500,000 was defined as a “Small city”. AMU: Antimicrobial use. PID: Patients/1000 inhabitants/day.

**Table 4 antibiotics-11-00763-t004:** The concordance rate of changes in AMU of the secondary medical areas in the Kyoto prefecture and previously reported AMUs.

	Japan				Kyoto			
	<15 Years	15–64 Years	>64 Years	All Ages	<15 Years	15–64 Years	>64 Years	All Ages
**Third-generation** **cephalosporins**	83.3%	33.3%	66.7%	25.0%	83.3%	33.3%	66.7%	25.0%
**Quinolones**	83.3%	41.7%	41.7%	33.3%	83.3%	41.7%	41.7%	33.3%
**Macrolides**	75.0%	75.0%	25.0%	66.7%	75.0%	75.0%	75.0%	66.7%
**Total**	66.7%	66.7%	75.0%	50.0%	66.7%	66.7%	75.0%	50.0%

The gray areas show a concordance rate of 50% or less. AMU: Antimicrobial use.

## Data Availability

The Ministry of Health, Labour and Welfare of Japan has placed strict legal restrictions on the release or sharing of the National Database of Health Insurance Claims and Specific Health Checkups of Japan. As a result, these data are not publicly available. But researchers can apply for the use of such data to the Ministry of Health, Labour and Welfare of Japan if they pass the qualification examination (phone number: +81-50-5546-9167).

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
