# Peer review of "Search for Indexes to Evaluate Trends in Antibiotic Use in the Sub-Prefectural Regions Using the National Database of Health Insurance Claims and Specific Health Checkups of Japan"

_antibiotics, 2022, doi:10.3390/antibiotics11060763_

Round 1
Reviewer 1 Report
Some points need to be addressed:
- In the abstract, it is not clear whta is the aim of this paper. Revise.
- Lines 168-170. "The distribution of ΔPID for inpatients and outpatients in the Kyoto prefecture is shown in Figure 1. The ΔPID of inpatients was concen trated around 0, while the ΔPID of outpatients was widely dispersed" In my opinion this point should be discuss more in discussion section. Which of the two ΔPID contributes more to the problem reported in this paper ?
- Line 228. "Previous reports indicated that AMU variation differs between Japan and prefectures 228 [7]." do the authors know why? improve
- Line 268-270: "The method used in this 268 study can be applied not only to Japan but also to other regions of the world... " Can the authors include in the discussion how to apply this method with some guidelines?
Author Response
Response to Reviewer #1
We would like to thank you for your review of this paper and for your helpful suggestions. Our corrections and comments are as follows. Corrections and additions are indicated in red in the manuscript. We would appreciate it if you could confirm these corrections and additions.
Comment 1
In the abstract, it is not clear what is the aim of this paper. Revise.
Response 1: Thank you very much for your comments. As the reviewer pointed out, the aim of this study was not clear in the abstract. We have changed the abstract.
Delete(s): Abstract (Page 1, Line20~21) National values for antimicrobial use (AMU) have been identified for each country, but sub-prefectural AMUs have not been identified.
Change(s): Abstract (Page 1, Line20~22) The evaluation indexes of antimicrobial use (AMU) in sub-prefectural regions have not been established because these regional units are susceptible to the effects of population inflows and outflows. |
Comment 2
Lines 168-170. "The distribution of ΔPID for inpatients and outpatients in the Kyoto prefecture is shown in Figure 1. The ΔPID of inpatients was concentrated around 0, while the ΔPID of outpatients was widely dispersed" In my opinion this point should be discuss more in discussion section. Which of the two ΔPID contributes more to the problem reported in this paper?
Response 2: Thank you very much for your comments. As the reviewer pointed out, the amplitude of ΔPID in outpatients was not adequately described. In this study, the ΔPIDs of both inpatients and outpatients are important. We added the discussion.
Change(s): Discussion (Page 7, Line216~220) ~suggesting that the number of patients using oral antibiotics in hospitalized patients scarcely changed. On the other hand, because the ΔPID in outpatients had been dispersed, it was found that outpatients had variability in oral antibiotic use. Therefore, it was considered necessary to take appropriate AMR measures separately for outpatients and inpatients. |
Comment 3
Line 228. "Previous reports indicated that AMU variation differs between Japan and prefectures 228 [7]." do the authors know why? Improve
Response 3: Thank you very much for your comments. We have added the reason why AMU variation differs between Japan and prefectures.
Change(s): Discussion (Page 7, Line233~234) Previous reports indicated that AMU variation differs between Japan and prefectures [7]. As one of these factors, the inflow and outflow of the prefectural population may have affected the AMU [15]. |
Comment 4
Line 268-270: "The method used in this 268 study can be applied not only to Japan but also to other regions of the world... " Can the authors include in the discussion how to apply this method with some guidelines?
Response 4: Thank you very much for your comments. In this study, the PID for each year was calculated and the difference was evaluated. We believe that this method can be applied not only to the PID but also to other AMU indexes. Therefore, we have added the following text.
Change(s): Discussion (Page 7, Line248~252) Evaluation using the difference in AMU during the study period is a useful method because it can be applied in the evaluation index of AMU other than PID, and can reflect the selection pressure of antibiotics after eliminating the effects of population inflows and outflows. In addition, this method makes it possible to evaluate whether the reduction in AMU has reached the goals of the guidelines in each country. |
Reviewer 2 Report
Kanako Mizuno and colleagues reported the antibiotics usage by population in Japan using the National Database of Health Insurance Claims and Specific Health Checkups. This study appears to be interesting and in my opinion, the same approach cannot be used in developing or resource limited countries. This approach may be applied in developed countries. The authors used appropriate methodologies to discuss their findings substantially. However, Authors may explain why they choose between 2013 and 2016, which seems to me one decade ago. And today we are on the blink over many available antibiotics. I would suggest authors to explain a bit more on this for the interest of readers.
Author Response
Response to Reviewer #2
We would like to thank you for your review of this paper and for your helpful suggestions. Our comments on your suggestions are as follows. We would appreciate it if you could confirm these corrections and additions.
Comment 1
Kanako Mizuno and colleagues reported the antibiotics usage by population in Japan using the National Database of Health Insurance Claims and Specific Health Checkups. This study appears to be interesting and in my opinion, the same approach cannot be used in developing or resource limited countries. This approach may be applied in developed countries. The authors used appropriate methodologies to discuss their findings substantially. However, Authors may explain why they choose between 2013 and 2016, which seems to me one decade ago. And today we are on the blink over many available antibiotics. I would suggest authors to explain a bit more on this for the interest of readers.
Response 1: Thank you very much for your comments. Other reviewers also commented on the application of this method using ΔPID to other countries. We have added discussion. NDB is strictly controlled by the country. In the application for the use of NDB in this study, it was available for four years, from 2013 to 2016. The aim of this study was to take the difference in AMU calculated in each year and define it as a new evaluation index of AMU. Therefore, the most recent data (2016) and the oldest data (2013) were used in this study.
Change(s): Discussion (Page 7, Line248~252) Evaluation using the difference in AMU during the study period is a useful method because it can be applied in the evaluation index of AMU other than PID, and can reflect the selection pressure of antibiotics after eliminating the effects of population inflows and outflows. In addition, this method makes it possible to evaluate whether the reduction in AMU has reached the goals of the guidelines in each country. |